# The Effect of Grain Size on Hyperspectral Polarization Data of Particulate Material

Rachel M. Golding, Christopher S. Lapszynski, Charles M. Bachmann * and Chris H. Lee

Chester F. Carlson Center for Imaging Science, Rochester Institute of Technology, Rochester, NY 14623-5603, USA
* Correspondence: cmbpci@rit.edu

**Abstract:** Polarization provides useful quantitative information about scattering surfaces. In hyperspectral remote sensing of natural surfaces composed of granular materials, there are relatively few studies of polarization. Most earlier remote sensing studies of polarization have been based on multi-spectral data, and the majority focused on the negative branch of polarization, which typically appears at phase angles less than 20 degrees, using models with limited accuracy. Models of the positive branch have also shown limitations, particularly at longer phase angles. We review these earlier studies by Hapke and Shkuratov and present the results of our laboratory study using hyperspectral polarization imagery of particulate surfaces. Although the linear polarization ratio is typically a nonlinear function of phase angle, our results show that in an approximately linear region of the polarization curve, there is a correlation between the slope of the linear polarization ratio and the average grain size.

**Keywords:** hyperspectral imagery; polarization; grain size; mean ray path

## 1. Introduction

In remote sensing, on which applications from weather forecasting to environmental studies rely, understanding the interaction between light and different materials offers useful information about the underlying properties of each material. Central to this understanding is the concept of polarization, since it offers potentially significant information useful for a number of applications, for example by providing a means of directly estimating the index of refraction [1]. In the context of hyperspectral imaging, there are a relatively limited number of past studies that have treated polarization in remote sensing analyses. Most of these have focused on remote sensing of the water column, atmospheric correction, or snow [2–5]. There have been only a few quantitative radiative transfer models used in the analysis of polarimetric hyperspectral data [6]. For remote sensing of granular materials (sediments, planetary regoliths), attempts to create a quantitative model for polarization have had only limited success [7–9] and by and large, these analyses have typically been applied to multi-spectral data.

Studies of the interactions of polarized light with surfaces or interfaces have a long history dating back to the early work of Fresnel [10]. In astronomy, historical roots can be traced back to planetary observations of the moon and planets by Arago (1811) and Lyot [7,11]. From a practical perspective, polarization ratios provide both useful information and the added advantage that absolute detector calibration is not required. In this work, we focus on the linear polarization ratio in the context of a hyperspectral imaging system and explore the effects of particulate grain size on the response of the linear polarization ratio as a function of phase angle. Our emphasis is on particulate media where the grain size relative to the wavelength of incident light can be considered to be in the geometric optics regime where radiative transfer models are appropriate. Models for smaller particle sizes have been considered from the perspective of Mie theory [12].

Between phase angles of 0 and 20 degrees, in some particulate media, the linear polarization ratio is negative, a region of the linear polarization curve referred to as the

negative branch, and the much larger positive branch extends from 20 to 180 degrees. Due to the complexity of modeling complex scattering interactions in terms of polarization, theoretical studies of polarization have dealt with the positive and negative branches separately [7,8]. The linear polarization ratio is defined in terms of a radiance ratio of the difference and sum of two orthogonal polarization states as defined in Equation (1):

$$P = \frac{I_\perp - I_\parallel}{I_\perp + I_\parallel} \tag{1}$$

where $I_\perp$ and $I_\parallel$ represent the perpendicular and parallel components of the radiance.

When a negative polarization branch exists, the typical minimum value in the negative branch is relatively small, $\sim-0.01$, at a phase angle below $20°$ and the maximum value on the positive side is around 0.1 at approximately $100–110°$ [7]. Influences on polarization come from varying forms of scattering, as explained in Figure 1.

- **Surface scattering from particles,** will tend to produce light that is **positively polarized** (Fresnel Equations)

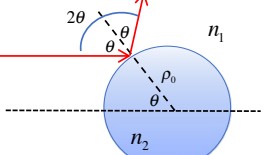

- **Volume scattering from within the particle** will tend to produce light that is **negatively polarized from refraction** on exiting the particle (Fresnel equations)

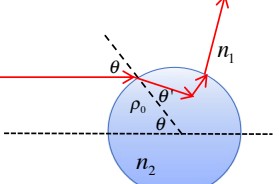

- **Multiple scattering:** for small angles, coherent backscatter produces negatively polarized light; at larger angles, leads to unpolarized light

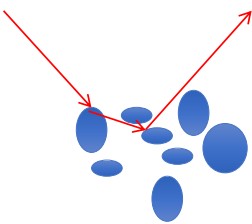

- **Diffraction: absent in a close-packed medium** since interference, between waves passing through particles with those just outside, is blocked in a close packed medium by the presence of other particles in the near field

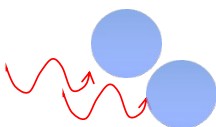

**Figure 1.** Scattering influences on the observed polarization in granular media. In close-packed media, principle contributions stem from: (**top**) surface scattering, (**upper middle**) volume scattering within particles, (**lower middle**) multiple scattering between particles. (**Bottom**) Diffraction effects can generally be ignored in the far field.

As Figure 1 illustrates, in a particulate medium, these observations stem from contributions due to surface scattering from particles (positively polarized light), volume scattering within particles (negatively polarized light), and multiple scattering between particles (negatively polarized light at small angles due to coherent backscatter effects [7,13], but otherwise unpolarized at larger angles). The coherent backscatter opposition effect (CBOE) has been well studied as one mechanism for the increased brightness at small phase angles observed in scattering from granular media, specifically at very small phase angles (typically $\leq 2°$), and analysis and comparison with experiment suggest that low-order

multiple scattering contributions (≤4 scattering events) typically contribute most at these small phase angles [7,13]. For light originating from a larger number of scattering events, the trend is toward randomization of the polarization, which in the aggregate leads to an unpolarized contribution. Figure 1 also illustrates that, as was observed previously, in a close-packed medium, diffraction effects can generally be ignored in the far field because interference in the near field is blocked by the presence of nearby particles [7].

Hapke developed a polarization model based on their Isotropic Multiple Scattering Approximation (IMSA) radiative transfer model [7], which will be discussed further in the next section. However, his model only focuses on the positive branch of polarization and does not agree with observational data at larger phase angles. This may be due in part to the assumptions made in model development, such as the neglect of negative polarization stemming from light transmitted through particles, the equal division of multiple scattering between the parallel and perpendicular radiance components, and the chosen isotropic form of multiple scattering used [7].

Prior to 1994, Shkuratov conceived several studies to model the negative branch in terms of different scattering phenomena. Shkuratov eventually summarized these models dividing them into 4 categories [8]. The first three are multiple reflection, refraction, and diffraction which were based on Lyot's three hypotheses [8]. The final and most promising category used models of coherent backscatter.

This study examines the effect of grain size distribution on observed polarimetric hyperspectral reflectance data. The observed link between these provides insight that may help to improve models of polarization overall through analysis of physical parameters which impact polarization, allowing for better inversion and retrieval of these parameters. In our study, we focus on granular materials which may have both surface and volume scattering. In these materials, as noted above, the scattering mechanisms exhibited by granular materials, which appear in Figure 1, include multiple effects that contribute to the polarization. This contrasts with materials that exhibit primarily strong surface scattering and are better described by the Fresnel equations. While the impact of grain size on polarimetric data has been studied before by those mentioned above, here this relationship is studied using hyperspectral polarimetric data, giving much more insight into the wavelength dependence of polarimetric imagery of particulate surfaces. We also derive specific relationships between grain size and observed changes in an approximately linear region of the positive polarization branch.

## 2. Background

### 2.1. Hapke's Model for the Positive Polarization Branch

Hapke's solution to the radiative transfer equation, with the assumption of isotropic multiple scattering, is the basis for a model of the positive branch of polarization that matches observations for moderate phase angles [7]. Assuming phase angles greater than 20° and therefore neglecting opposition effects, the radiance solution is [7]:

$$I(i,e,g) = JK\frac{w}{4\pi}\frac{\mu_{0e}}{\mu_{0e} + \mu_e}\left(\overbrace{p(g)}^{\text{Single scattering}} + \underbrace{\left(H\left(\frac{\mu_{0e}}{K}\right)H\left(\frac{\mu_e}{K}\right) - 1\right)}_{\text{multiple scattering}}\right)\overbrace{S(i,e,g)}^{\substack{\text{macroscopic roughness}\\\text{correction}}} \tag{2}$$

where $J$ is the incident irradiance, $K$ is a nonlinear porosity function depending on the filling factor $\phi$ of the granular material, $w$ is the wavelength-dependent single scattering albedo, $p(g)$ is the single scattering phase function, dependent on phase angle $g$, $H$ is the Ambartsumian-Chandrasekhar $H$-function [14] describing multiple scattering, $S(i,e,g)$ is Hapke's correction for macroscopic roughness [7,15], $\mu_{0e}$ and $\mu_e$ are the effective incident and observation direction cosines of the locally tilted surface if the surface is rough and otherwise reduce to the incident and observation direction cosines of a flat surface in the absence of macroscopic roughness, and $i$ and $e$ denote the incident and observation zenith angles in a locally flat coordinate system [7]. Further details of the derivation can be found

in [7]. More recent modeling efforts have provided more accurate models for macroscopic roughness [16]; however, as we will see below, the dependence on the macroscopic roughness factor $S(i, e, g)$ is eliminated in the linear polarization ratio, although the direction cosines $\mu_0 e$ and $\mu_e$ that appear in the $H$-functions would be the effective direction cosines for a surface that is not smooth if macroscopic roughness is present. The multiple scattering term shown in Equation (2) is an isotropic form. As explained earlier, this equation can be further broken down into perpendicular and parallel components by assuming that the multiple scattering term is divided equally between the two, although this assumption is a potential reason for inconsistencies between the model and observations. The model parallel and perpendicular components are then [7]:

$$
\begin{aligned}
I_\perp(\mu_i, \mu_e, g) &= JK \frac{w}{4\pi} \frac{\mu_{0e}}{\mu_{0e} + \mu_e} \Big( [p(g)]_\perp \\
&+ \frac{1}{2}(H(\frac{\mu_{0e}}{K})H(\frac{\mu_e}{K}) - 1) \Big) S(i, e, g)
\end{aligned}
\tag{3}
$$

$$
\begin{aligned}
I_\parallel(\mu_i, \mu_e, g) &= JK \frac{w}{4\pi} \frac{\mu_{0e}}{\mu_{0e} + \mu_e} \Big( [p(g)]_\parallel \\
&+ \frac{1}{2}(H(\frac{\mu_{0e}}{K})H(\frac{\mu_e}{K}) - 1) \Big) S(i, e, g)
\end{aligned}
\tag{4}
$$

Combining these expression with Equation (1), one obtains the expression:

$$
P(\mu_i, \mu_e, g) = \frac{\omega \Delta p(g)}{\omega p(g) + \omega(H(\frac{\mu_{0e}}{K})H(\frac{\mu_e}{K}) - 1)}
\tag{5}
$$

where $w\Delta g(g) = w[p(g)]_\perp - w[p(g)]$ depends on Fresnel coefficients, $R_\perp$ and $R_{parallel}$, and an unpolarized residual term, $\frac{w - S_e}{2}$, expressed in terms of the single scattering albedo:

$$
w[p(g)]_\perp = R_\perp(\frac{g}{2}) + \frac{w - S_e}{2}
\tag{6}
$$

$$
w[p(g)]_\parallel = R_\parallel(\frac{g}{2}) + \frac{w - S_e}{2}
\tag{7}
$$

where $S_e$ is the external surface reflection coefficient. The final resulting expression for polarization becomes [7]:

$$
P(\mu_i, \mu_e, g) = \frac{R_\perp(\frac{g}{2}) - R_\parallel(\frac{g}{2})}{2(R_\perp(\frac{g}{2}) + R_\parallel(\frac{g}{2})) + \omega - S_e + \omega(H(\frac{\mu_0}{K})H(\frac{\mu_e}{K}) - 1)}
\tag{8}
$$

Further details can be found in [7].

While the model is conceptually straightforward, Hapke found that the model fails to predict the polarization maximum in observed data as well as the angle at which it occurs. This appears to be a limitation of the use of the Fresnel coefficients [7]. The difference between Fresnel coefficients in the numerator, and subsequently the polarization model as a whole, peaks at higher phase angles than in observational data [7]. This is increasingly true as the unpolarized residual term, $w - S_e$, increases. Furthermore, this model does not accurately predict the polarization at phase angles greater than 80°, likely due to the approximate isotropic form of multiple scattering within the model. There is strong evidence that multiple scattering is not isotropic and is not evenly divided between the perpendicular and parallel orientations to the scattering plane. Furthermore, it may be necessary to insert the IMSA model into a Stokes' matrix representation rather than reducing polarization information used to just two orientations. In addition, Equation (8) does not explicitly include particle size as a free parameter.

### 2.2. Models for the Negative Polarization Branch

Current models for the negative branch of polarization do not completely describe observations [7,8]. In addition to influencing the shape of the negative branch at small phase angles, negative polarization also likely plays a significant role at higher phase angles due to contributions from refracted light that enters particles but also escapes the particle in forward scattered directions. As Hapke has pointed out, such light will be negatively polarized [7]. At small phase angles near opposition, contributions likely stem from three sources: the coherent backscatter opposition effect (CBOE) [7,13], polarization opposition effect (POE) [7], and broad negative polarization (BNP) [7]. It is important to emphasize that these effects are expected for very fine particles [7] where the geometric optics model described earlier for the positive branch may not apply; however, even particles in the geometric optics regime may exhibit a negative polarization branch [7,8].

The negative branch is often bimodal, and the POE is the likely cause of the smaller-angle negative peak while BNP, which refers to the second peak, is not well understood [7]. Even the origin of the bimodal trend is uncertain, as the strength of the POE can be low. Quantitative understanding of the CBOE is somewhat developed, but virtually nonexistent for POE and the BNP. Even a qualitative understanding of BNP is lacking, other than its angular location very close to the angular region where the shadow-hiding opposition effect (increased reflectance) occurs at angles less than 20° [7].

Several historical models were reviewed by Shkuratov [8]; however, since the majority of our data does not exhibit a negative polarization branch, likely due to larger particles sizes and lower bulk density, we limit our discussion here and refer the reader to Shkuratov's earlier work [8] for further details on the experiments that he undertook to explore the limitations of these earlier models. His most successful models were based on coherent backscatter, which achieved better quantitative agreement with data. Among these, the more promising models were: (a) a second-order Fresnel reflection model with an exponential probability of light propagation between two scatterers and (b) a second-order point scatterer model based on a medium of small particles bounded by a plane and characterized by a single scattering albedo, a Rayleigh scattering phase function, and a ratio of particle radius to wavelength [8]. The first model predicts the wavelength dependence of polarization well, but would be difficult to apply practically to remote sensing imagery, while the second model, required significant simplification to match experimental data, and its use of a Rayleigh phase function limits applicability to very small particles. Other models, such as the discrete dipole approximation (DDA) [17], have been used to investigate, for example, the relationship between refractive index and the polarization minimum and its associated phase angle for irregular agglomerate particles, determining that the ratio of the phase angle to the cube of the real part of the refractive index remains constant [18]. Similarly, other studies using the same model have investigated the relationship between albedo and the polarization minimum [9]. Meanwhile, other works have examined the relationship between the polarization minimum and associated phase angle with variables such as particle size, index of refraction, and particle density and found that the polarization minimum is related to grain size, while also being influenced by these other variables [19].

### 2.3. Other Historical Models of Polarization

For small particles, a series of experimental studies by Kerker also explored the relationship of grain size and polarimetry [12], typically for small particles on a scale similar to the wavelength, such as aerosols and particles dissolved in solutions, where Mie scattering theory applies. Highly accurate experimental models were based on zero-order logarithmic distribution (ZOLD) functions [20] and incorporated multi-angular measurements to optimize models [12,21]. Similarly, other studies in the 1970s analyzed the wavelength dependence of polarization by observing interstellar grains [22]. These models assumed cylinders of constant length, a size distribution similar to the Oort-van de Hulst distribution, with refractive index m = 1.33 and orientation defined by the Davis-Greenstein

mechanism [23] with orientation parameter 0.1–0.4. Their empirical polarization model form was

$$P/P_{max} = exp[-Kln^2(\lambda_{max}/\lambda)] \tag{9}$$

where $\lambda_{max}$ was the polarization maximum average over the observational period. Their empirical model fit observations well and matched classical models over a wide range of $\lambda_{max}$ for values of $\lambda$ between 0.22 and 2.2 µm [22]. A related grain size retrieval model was based on their observation that wavelength dependence of polarization played a more significant role than the absolute value of polarization, and that the nature of the grains of a surface could be narrowed down through the wavelength dependence of polarization [24].

Sun et al. compared the polarization of black soil to that of a particular type of sand (S2) [6]. Their experimental data covered wavelengths ranging from 300 to 2500 nm, and they grouped their sample particles into diameter ranges: 300 µm or less, 300–450 µm, 450 µm, 450–900 µm, and 900 µm. Results showed that the black soil had more prominent positive and negative polarization peaks than the sand, for data in the spectral range 350–2300 nm. The results also suggested that negative polarization is more prominent for larger grain sizes than smaller ones, and that this relationship is more obvious with low-reflectance materials than high-reflectance materials [6], which is the opposite of what we found for the materials that we examined in this work.

## 3. Objectives

In this study, we analyze the impact of average grain size in granular materials on the observed polarization of hyperspectral imagery and use the observed trends to derive empirical relationships between grain size and polarization. Other parameters potentially influencing the polarization of spectral imagery include moisture level, material composition, and albedo. However, in this study, we focus on wavelength-dependent observations of grain size in materials of the same composition, observed using a hyperspectral imaging system [25]. Some progress was made by Shkuratov in identifying parameters that affect the negative polarization branch, but the nature of the relationship is still not well understood, and in this study, the materials analyzed did not exhibit a significant negative branch, likely due to the fact that the materials were representative of the geometric optics scattering regime in the observed wavelength range (400–100 nm) and were lower density having been sifted; Hapke and others previously found that bulk density is a determining factor for the presence of a negative branch, with more densely packed granular materials showing a strong negative branch and loosely packed materials exhibiting a minimal negative branch [7]. Many previous studies concentrated on materials where particle sizes were significantly smaller relative to the observation wavelengths, likely placing these materials in either the resonance regime or Rayleigh regime.

## 4. Methods

### 4.1. Materials and Equipment

The samples analyzed in this experiment were olivine sediment with grain sizes ranging from 50 to 800 µm provided by Washington Mills, and four samples provided by AGSCO, including an olivine sample with grain sizes ranging from 60 to 1000 µm, a silica sample with grain sizes from 250 to 850 µm, and two samples of nepheline sand, one with grain sizes ranging from 250 to 850 µm and the other from 1 to 5 µm. Because the grain sizes were comparable to the wavelength range of our hyperspectral imaging system (0.400–1.000 µm), the latter sample fell within the resonance, or Mie scattering, regime, while the others, with significantly larger grain sizes, correspond to the geometric optics regime. Radiance measurements were also recorded from a standard $Spectralon^{TM}$ white reference panel, but were not required to determine the linear polarization ratio used in this study. The samples corresponding to the geometric optics regime were sifted into multiple groups for grain size analysis. This provided a larger number of grain size categories to explore better the relationship between polarization and grain size. The specific grain size groupings appear in the Results section. We prepared the geometric optics regime samples

in a black opaque circular container with a 20.32 cm diameter and a cardboard insert placed in the bottom of the container, making the sample depth 1.9 cm deep. We prepared the resonance regime sample without the insert so that the sample depth was 3.8 cm. Figure 2 shows example photographs of each AGSCO-prepared sample. Figure 3 highlights the difference between the AGSCO 63–300 μm olivine and 600–1000 μm olivine samples; these samples correspond to the largest contrast in grain size across samples used in our study, and the Figure highlights the obvious difference in material appearance due to differences in grain size.

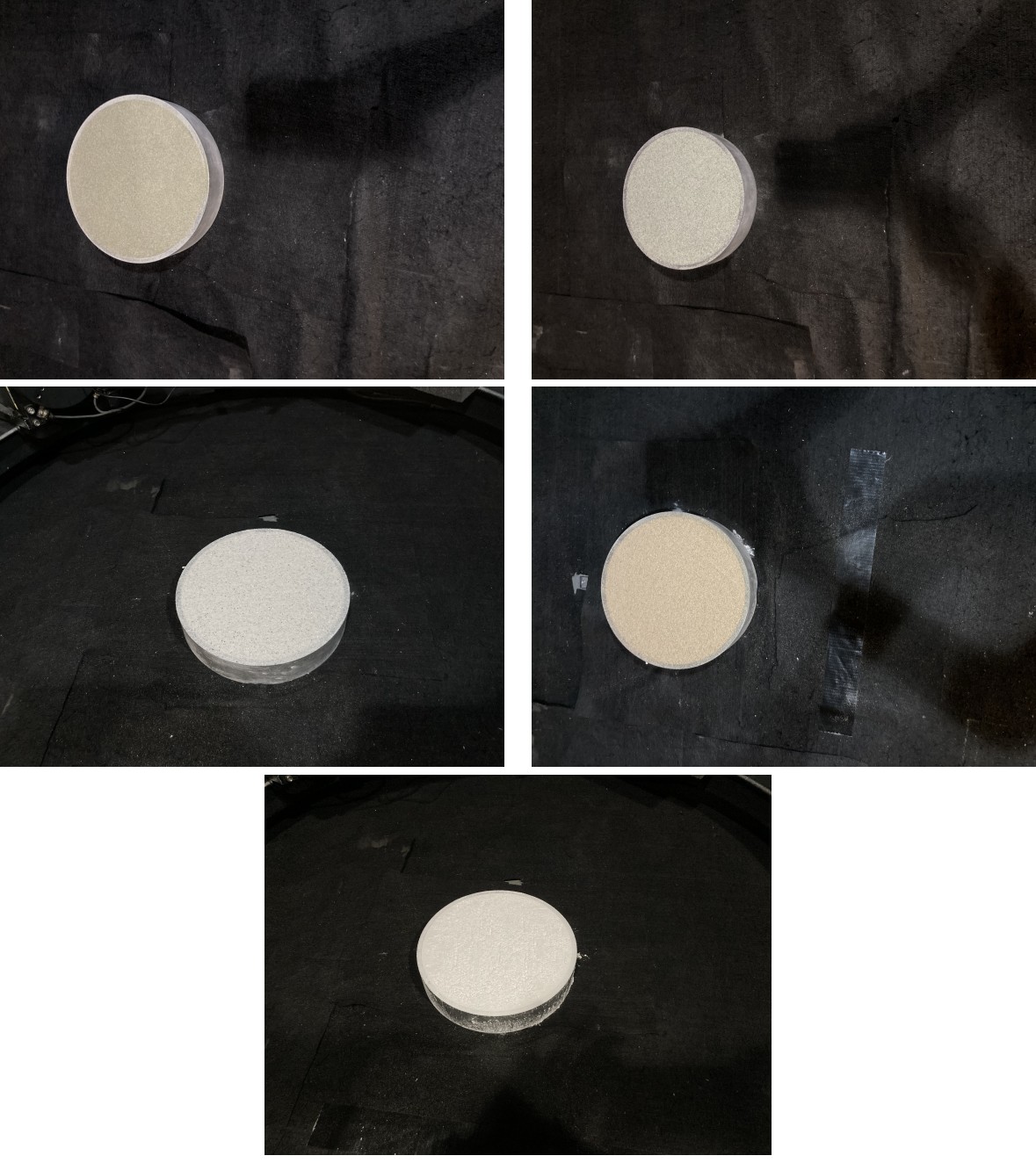

**Figure 2.** Photos of prepared samples of 250–425 μm olivine from Washington Mills (**top left**), and from AGSCO 500–600 μm olivine (**top right**), 500–600 μm nepheline (**middle left**), 500–600 μm silica (**middle, right**), and 1–5 μm nepheline (**bottom**).

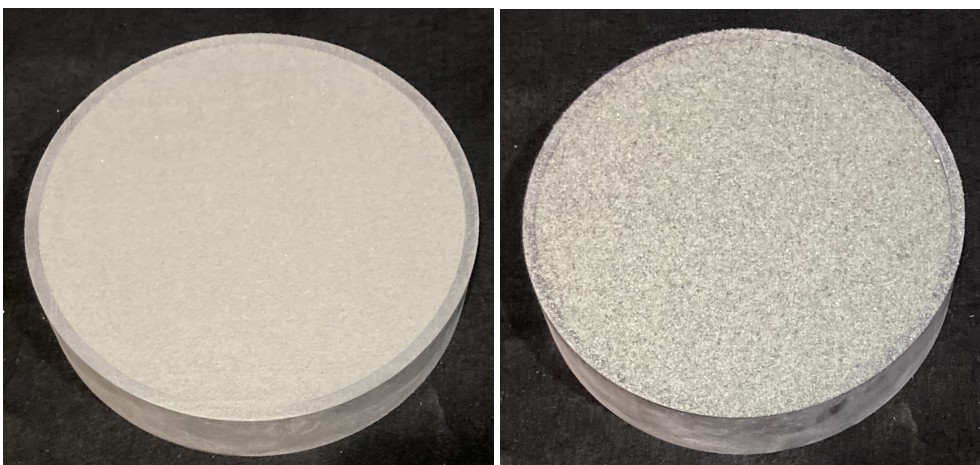

**Figure 3.** Photos of prepared samples of AGSCO 63–300 μm olivine (**left**) and 600–1000 μm olivine (**right**).

We collected hyperspectral imagery in our laboratory using a Headwall Hyperspec micro-HE (high efficiency) E-Series imaging system mounted on a General Dynamics pan-tilt unit [25]. A 50 mm polarizer was mounted at the front of the optical pathway [26]. The hyperspectral imaging system and polarizer both interface to a computer where their parameters can be managed. The polarizer is controlled through a stepper and DC motor controller and is mounted on a 360 deg rotation stage [27,28]. Within this configuration, the hyperspectral imaging system view geometry, polarizer angle, exposure, and field of view can be adjusted.

The hyperspectral imaging system collected imagery in 371 bands ranging from 400–1000 nm, sweeping from 15 to 30 deg below the horizontal. The Hyperspec is a pushbroom hyperspectral line scanner where the along-track spatial dimension is acquired by the motion of the imaging system. In an aircraft, this is generated by the platform motion, while in our system, the nodding of the pan-tilt unit produces the along-track spatial dimension [25]. This angular range placed the sample being analyzed in the center of the frame where it was also aligned with an unpolarized light source in the principal plane. For each data collection, we acquired hyperspectral imagery with the polarizer oriented at 0, 45, 90, and 135 degrees. The 0 and 90 deg orientations were used to calculate the linear polarization ratio using the raw digital count values of the two images, and the two oblique orientations were used as a test of the significance of adding those orientations associated with the other Stokes components. The light source is positioned on a transom attached to a rotation stage, which interfaces to a separate computer. To ensure that a wide range of phase angles could be measured, we rotated the illumination source via the rotation stage, changing the illumination zenith angle across a series of angular positions ranging between 25 and 135 deg with respect to the neutral position of the hyperspectral imaging system, with the light source increment being in 5 deg intervals. Figure 4 shows the lab setup from the point of view of our hyperspectral imaging system and from the side as well as the front of the hyperspectral imaging system with the polarizer attached.

A three-band image derived from one of the example hyperspectral images used in our study appears in Figure 5. In the Figure, approximately 3/4 of the output image width has been cropped, evenly on each side of the image, to remove the black background of our laboratory environment. The red, green, and blue bands appearing in the image represent wavelengths at 650.00 nm, 550.0231 nm, and 451.097 nm, respectively.

As shown, only a small portion of the image includes the sample itself, which necessitated masking of the background portions of the image. Our surrounding laboratory environment has black walls, ceiling panels, and flooring, and surrounding lights were shut off to minimize background light reaching the hyperspectral imaging system.

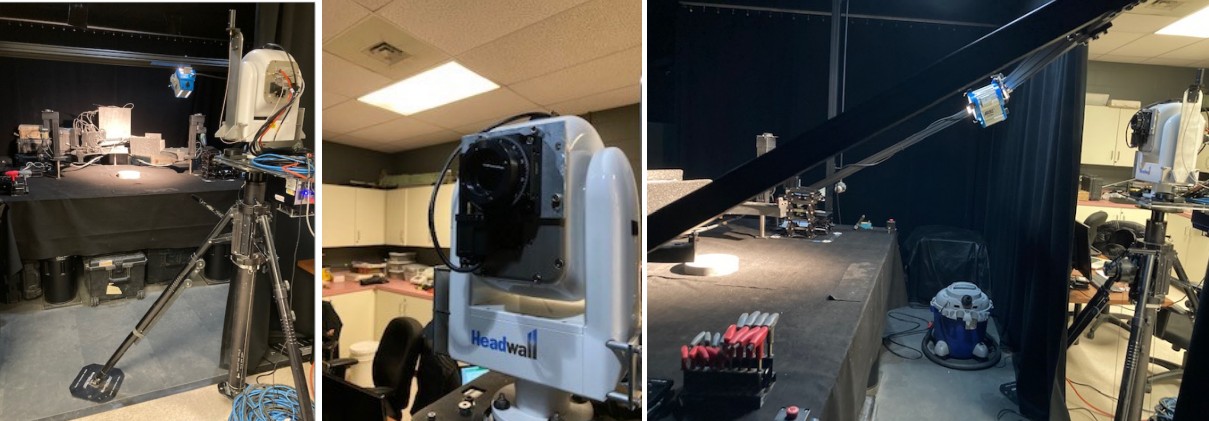

**Figure 4.** (**Left**) Experimental Setup from point of view of the hyperspectral imaging system, (**middle**) the hyperspectral imaging system with the polarizer and rotation stage attached, (**right**) side view of the experimental configuration.

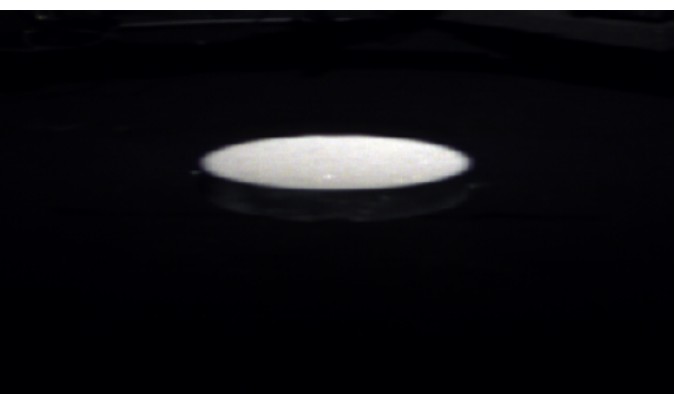

**Figure 5.** Example hyperspectral Raw DN output image for a 500–600 µm Nepheline sample with polarizer zenith angle of 0 deg and illumination zenith angle of 50 deg. Displayed RGB bands selected from the hyperspectral imagery are, respectively, band 156 (650.99 nm), band 94 (550.231 nm), and band 33 (451.097 nm).

Additionally, the hyperspectral imaging system was calibrated by us using a Lab-Sphere Helios integrating sphere [25], and these data were used to convert the raw digital numbers (DN) to radiance before calculating the polarization ratio.

### 4.2. Processing and Calculations

After collecting the data for each sample, a mask was developed in $ENVI^{TM}$ using a rectangular region of interest (ROI), which was manually drawn within the sample region of the image. Since the sample was not moved during each experiment, only one mask was needed for each data collection. The mask was specifically designed to exclude pixels close to the edge of the sample holder; however, the included portion contained a sufficient amount of data for analysis.

After creating the mask, the remaining processing and calculations were completed in Python. Images were read into Python, then the 0 and 90 deg polarizer orientations for each light angle were converted to radiance using the polarized calibration data. We acquired the calibration data using a 0.5 m HELIOS® 20″ integrating sphere from Labsphere with an integrated 3000 K quartz tungsten halogen light source. We imaged the open port of the sphere with the Headwall imaging spectrometer with the affixed linear polarization stage. Specifically, we acquired radiance measurements using the calibrated spectrometer attached to the integrating sphere while simultaneously imaging the exit port of the sphere using the Headwall imaging spectrometer and linear polarizer configuration. Ten different

illumination levels within the sphere were used to capture the full dynamic range of the Headwall imaging spectrometer at both 0° and 90° polarization angles. For our Headwall imaging system, the same exposure time of 100 ms used during the laboratory sample measurements was used for the integrating sphere calibration measurements. Using our calibration measurements, the digital number (DN) images of the particulate samples were then converted to calibrated radiance images, which were used to calculate the linear polarization ratio for each image pixel using Equation (1). The phase angle was calculated for each line of the image by subtracting the camera angle (which changes with the line number of the image) from the angle of the light source (which was changed for each set of four images at the aforementioned four polarizer orientations). Since the linear polarization ratio defined in Equation (1) is unitless, it was not theoretically necessary to convert raw DN values to radiance or reflectance; however, the use of calibrated radiance mitigates potential polarization bias, which could otherwise adversely affect the calculation of the linear polarization ratio in Equation (1). Tests were conducted comparing the results using the polarization ratio of Equation (1) to the results when using the 45 and 135 deg orientations as well in a Stokes analysis [29] where the polarization is defined as follows:

$$P = \frac{\sqrt{Q^2 + U^2}}{I} \tag{10}$$

where:

$$I = Image_\perp + Image_\parallel \tag{11}$$
$$Q = Image_\perp - Image_\parallel \tag{12}$$
$$U = Image_{45} - Image_{135} \tag{13}$$

Using the Stokes polarization that incorporated these additional polarization orientations did not significantly change the results. Therefore, the linear polarization ratio in Equation (1), using just two components, was sufficient for these samples.

After cropping out the background, the data were averaged along each line of the sample since the phase angle remains approximately constant along each line over the width of the sample holder. This step condensed the data significantly, improved the signal-to-noise ratio (SNR), and allowed for ease of analysis. Polynomial curves of varying orders were then fit to this averaged data to determine the most appropriate model based on the root mean squared error (RMSE). Finally, the slope of the linear region of the data was plotted against the average grain size of each measured sample. Further details of the analysis appear in the next section.

## 5. Results and Discussion

Figure 6 shows the linear polarization ratio for a representative wavelength (633 nm) for each sample measured, averaged along each line of the sample in the image data as described earlier, with one mean value in the plot for every row of the image after masking out the background. As shown, the olivine from Washington Mills displays a much more distinct separation and trend as a function of grain size than the other geometric-optics regime samples, including the AGSCO olivine. However, the AGSCO olivine does have narrower steps between grain size categories than the Washington Mills olivine, so the behavior might have been more similar if the samples could have been binned the same way, and we note that the separation in polarization between the largest and smallest grains is actually similar between the two olivine samples. The AGSCO olivine, however, does have more separation than the nepheline or silica. This may be due to a combination of differences in material properties and the specific grain size categories chosen. The Washington Mills olivine shows a much greater increase in the polarization maximum with increasing grain size than the other samples, for which the maximum increases more gradually. The reason for this will require further study. However, when we examine the best-fit polynomials to each curve in Figure 7, every sample among the

geometric optics regime-sized samples shows some separation between curves for the different grain sizes at phase angles of 45 deg or higher.

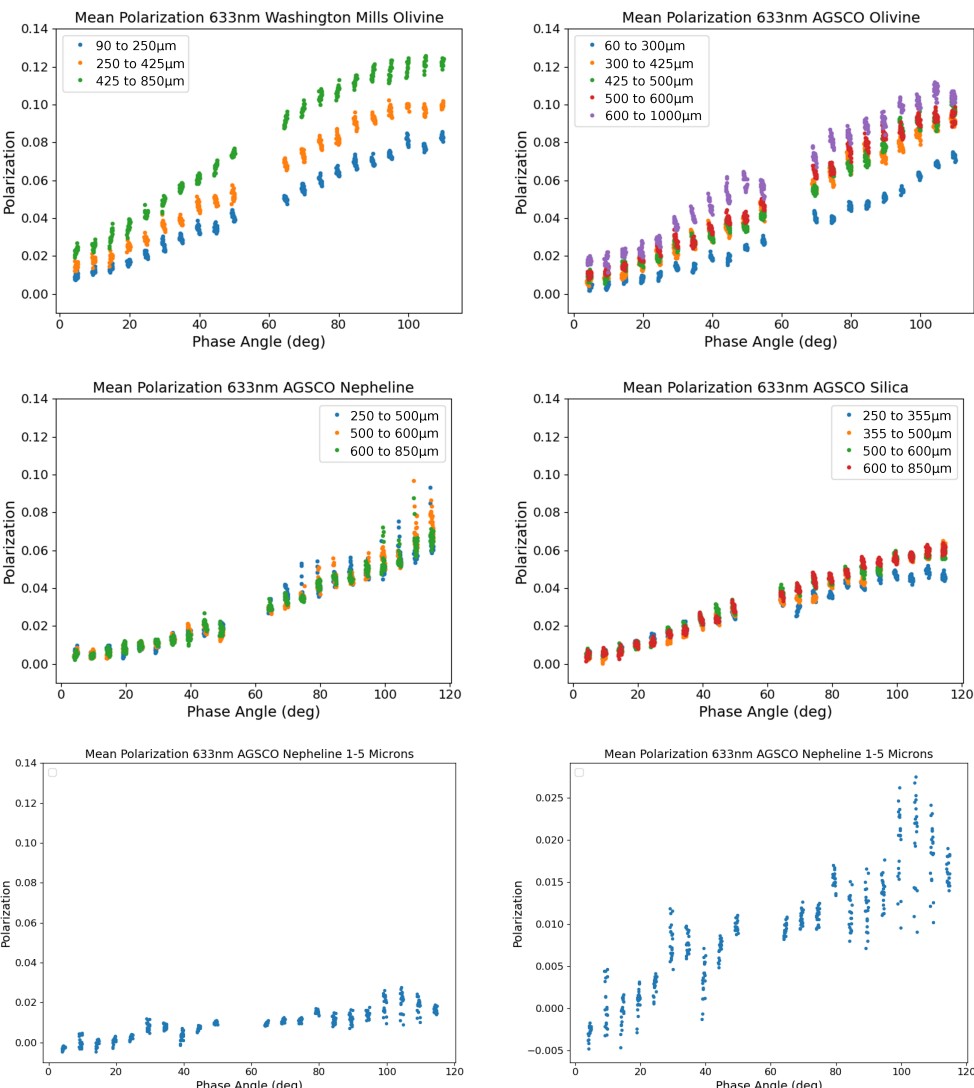

**Figure 6.** Average linear polarization of each hyperspectral image sample line for grain size subsets: (**top left**) Washington Mills olivine; (**top right**) AGSCO olivine; (**middle left**) AGSCO nepheline; (**middle right**) AGSCO silica; (**bottom left**) AGSCO resonance regime-sized nepheline on the same scale as other samples; (**bottom right**) AGSCO resonance-regime nepheline on a scale matched to the dynamic range of the data.

The final plots in Figures 6 and 7 show the nepheline sample with resonance regime-sized particles. Here, the particle size distribution is better described by Mie Scattering due to the relative size of the particles compared to the wavelength. Because of its distinct behavior, in Figure 6, a version of these data is shown with a zoomed-in scale to provide greater detail. A point of interest is that the curve for this sample is significantly flatter beyond a phase angle of 30 deg than for the other samples. None of the samples showed a negative polarization branch, except for the resonance regime-sized sample below a phase angle of 20°. This is in line with Shkuratov's results for resonance-sized samples. We return to this observation in our discussion of the results later in this section.

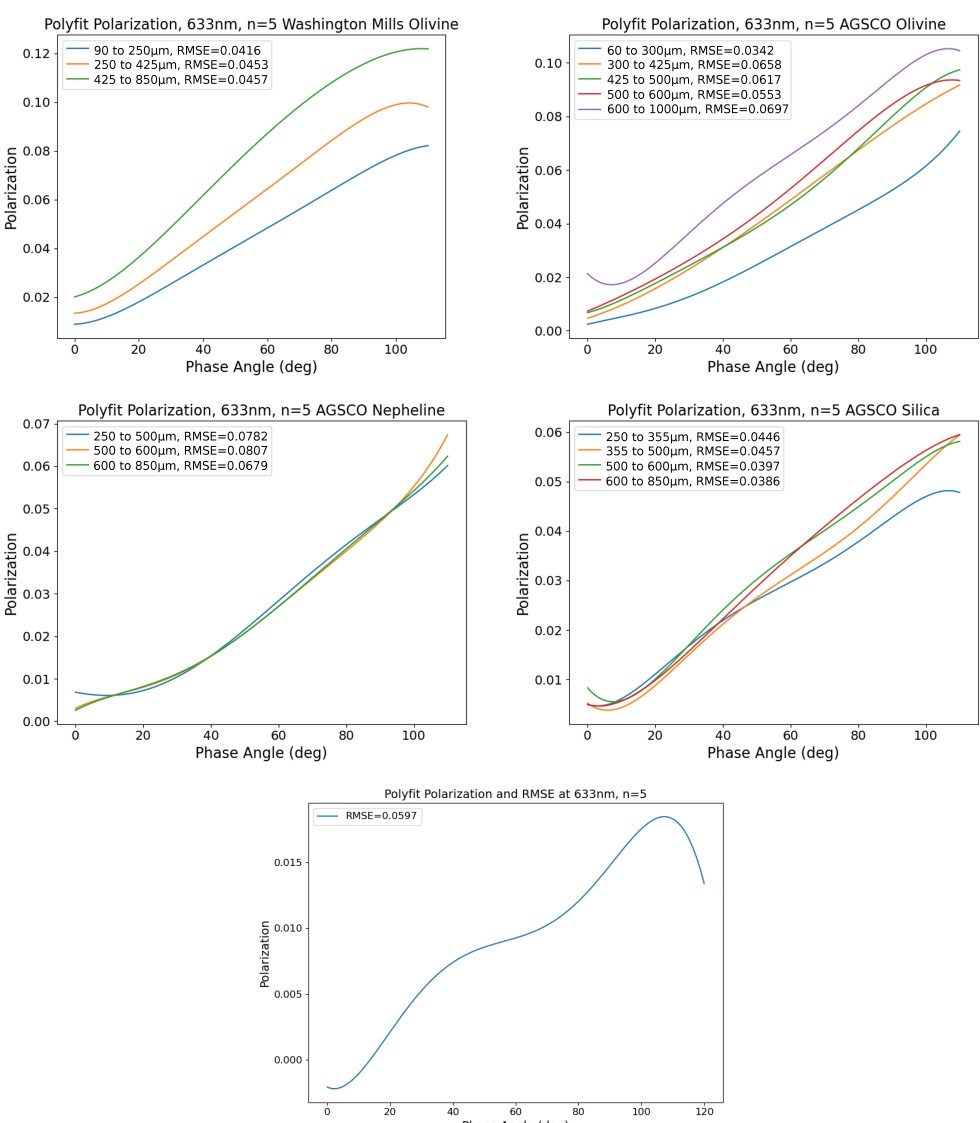

**Figure 7.** Fifth order polynomial fits to averaged linear polarization vs. phase angle with RMSE for binned samples of various grain sizes: (**top left**) Washington Mills olivine grain sizes between 90–250 μm, 250–425 μm, and 425–850 μm; (**top right**) AGSCO olivine between 60–300 μm, 300–425 μm, 425–500 μm, 500–600 μm, 600–1000 μm; (**middle left**) AGSCO nepheline between 250–500 μm, 500–600 μm, and 600–850 μm; (**middle right**) AGSCO silica between 250–355 μm, 355–500 μm, 500–600 μm, and 600–850 μm; and AGSCO nepheline from 1–5 μm (**bottom**).

Figure 7 shows a 5th-order polynomial fit to each of the datasets appearing in Figure 6. This offers an easier analysis of the behavior of the data in the macro scale, i.e., across the full range of phase angles. The general trend for the Washington Mills olivine remains consistent for the averaged data across the different grain size distributions; however, the slope increases significantly as grain size increases. In contrast, the other geometric-optics regime samples all have grain size-designated curves that intersect the other curves more than once. On the other hand, consistent with the Washington Mills olivine sample, the other geometric optics regime samples each have regions in which there is increasing polarization with grain size. The resonance sized sample is entirely distinct, even having a negative polarization branch and multiple significant slope changes.

Figure 7 also shows the RMSE of fifth-order polynomial fits to the data sets. The fits to linear polarization with phase angle did not exhibit significant wavelength dependence and had comparable RMSE provided wavelengths were sufficiently far from the edges of the

spectral range of the imaging system, avoiding lower signal-to-noise (SNR) regions of the imaging system. In our study, we found satisfactory data quality between 400 and 900 nm. Additionally, the necessary order of the model to achieve sufficient accuracy did not vary significantly with different grain size ranges within a sample. First, second and third-order fits were also considered, but had higher RMSE, while fifth-order fits captured the data most accurately. Fifth-order polynomial fits may not be sufficient to model samples displaying measurements taken at higher phase angles than the maximum range measured in this study (120 deg). Orders higher than 5 were tested and found to be unnecessary for the range of phase angles measured in our study, as these higher orders did not noticeably decrease the RMSE.

In addition to the trends already discussed, our measurements indicated a region of the polarization curve which was linear between phase angles of 20 and 80 deg for all of the samples considered. A linear fit at a wavelength of 633 nm for each sample appears in Figure 8, and for 425 and 850 nm in Figure 9, demonstrating the consistency of this trend across the wavelength range. Figure 8 shows these fits to be as accurate as the fifth order fits to the full-range data in Figure 6, and additional analysis not shown in this paper proved that the RMSE is not significantly changed between a linear fit and the fifth order curve fit within the 20–80 deg region, suggesting that this region is basically linear.

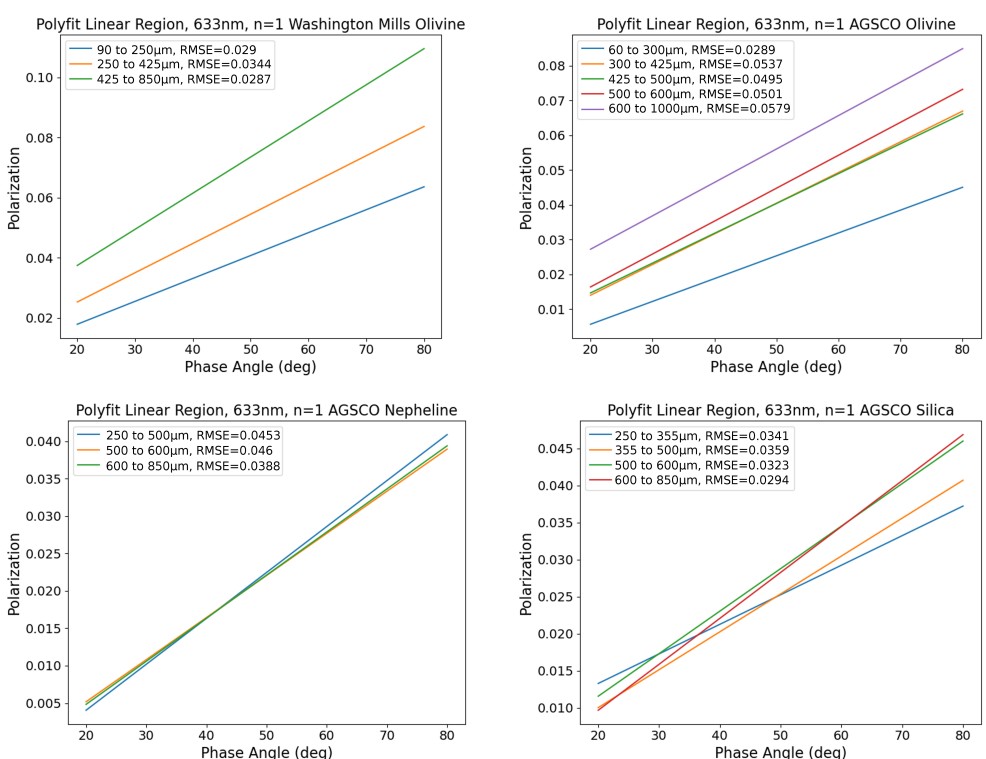

**Figure 8.** Linear fits and associated RMSE of the polarization ratio at $\lambda$ = 633 nm for phase angles between 20°–80° for various grain sizes of (**top left**) Washington Mills olivine, (**top right**) AGSCO olivine, (**bottom left**) AGSCO nepheline, and (**bottom right**) AGSCO Silica.

Figure 9 demonstrates that the RMSE of linear fits of data is still sufficient at 850 nm. For some of the samples, there is an increase in RMSE at 425 nm compared to the other two wavelengths shown, possibly because the 425 nm wavelength is too close to the lower end of the spectral range where SNR is lower. Nevertheless, a linear region was still observed at even those wavelengths at the extreme end of the spectral range (i.e., the RMSE even at 425 nm was not significantly worse for a linear vs. 5th order curve fit).

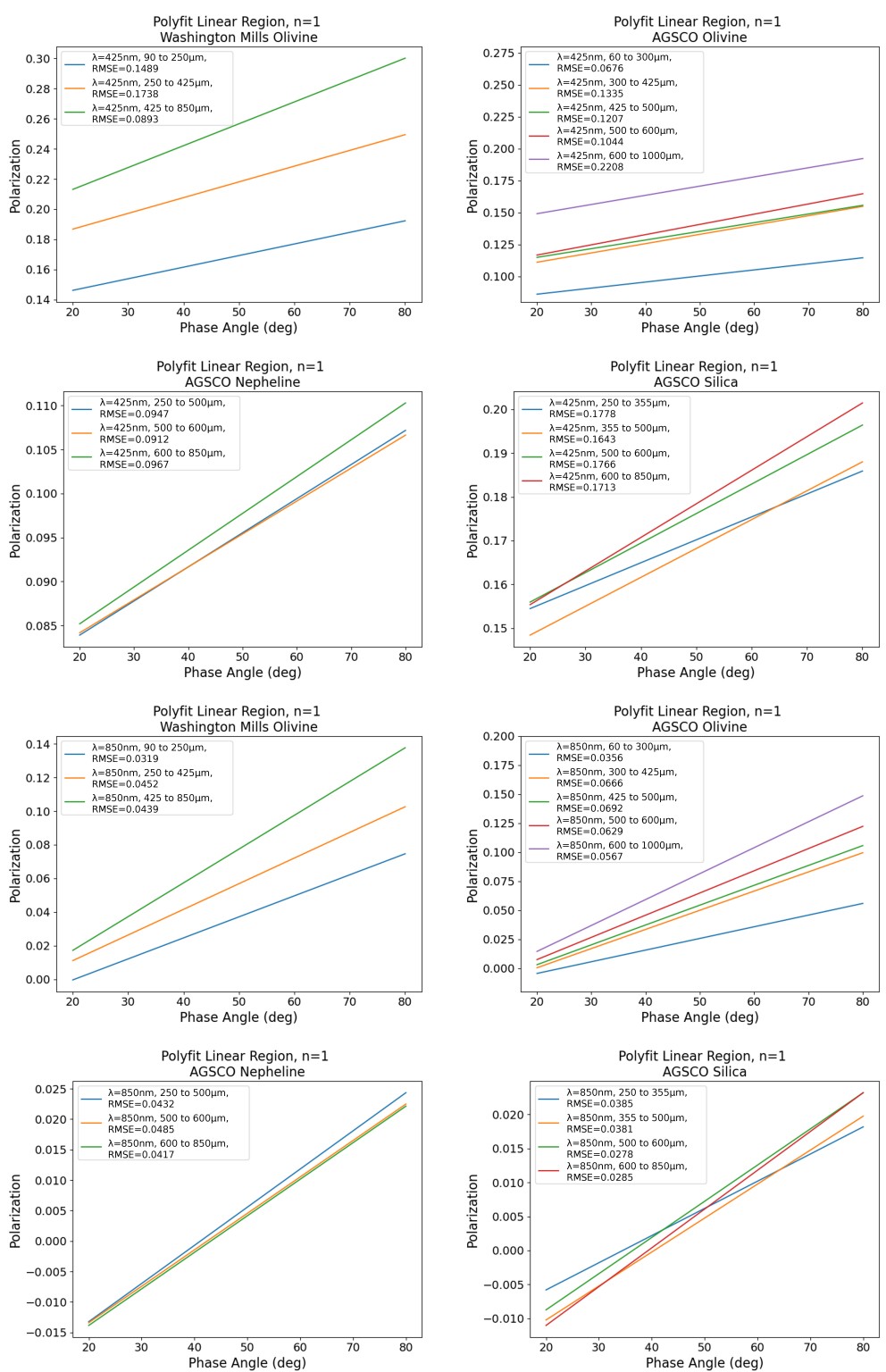

**Figure 9.** Linear fits and associated RMSE of the polarization curves for phase angles between 20°–80° for various grain sizes for (rows 1 and 2) λ = 425 nm and (rows 3 and 4) λ = 850 nm. (**left**, rows 1 and 3) Washington Mills olivine, (**right**, rows 1 and 3) AGSCO olivine, (**left**, rows 2 and 4) AGSCO nepheline, and (**right**, rows 2 and 4) AGSCO Silica.

We analyzed the impact of grain size, wavelength, and material type on the slope of the linear region. The results are shown for each respective material type in Table 1. The Table shows that, to some degree, this slope varies with all three of the aforementioned

properties. The one material which has a different trend, compared to the others, is the nepheline sample. As we discuss later, this material appears to be more strongly surface scattering than the other materials, which may have led to some of the apparent differences. We analyze these differences further below.

**Table 1.** Slope of Polarization for Each Sample in the Linear Region at Various Wavelengths.

| Material | Grain Size (μm) | 425 nm | 633 nm | 850 nm |
|---|---|---|---|---|
| Olivine (WM) | 90–250 | 0.000768 | 0.000762 | 0.001251 |
| Olivine (WM) | 250–425 | 0.001044 | 0.000973 | 0.001525 |
| Olivine (WM) | 425–850 | 0.001450 | 0.001202 | 0.002009 |
| Olovone (AGSCO) | 60–300 | 0.000475 | 0.000657 | 0.001005 |
| Olovone (AGSCO) | 300–425 | 0.000729 | 0.000882 | 0.001652 |
| Olovone (AGSCO) | 425–500 | 0.000679 | 0.000858 | 0.001710 |
| Olovone (AGSCO) | 500–600 | 0.000801 | 0.000947 | 0.001913 |
| Olovone (AGSCO) | 600–1000 | 0.000720 | 0.000961 | 0.002235 |
| Silica | 250–355 | 0.000524 | 0.000399 | 0.000399 |
| Silica | 355–500 | 0.000660 | 0.000511 | 0.000499 |
| Silica | 500–600 | 0.000674 | 0.000574 | 0.000532 |
| Silica | 600–850 | 0.000768 | 0.000620 | 0.000570 |
| Nepheline | 250–500 | 0.000388 | 0.000614 | 0.000627 |
| Nepheline | 500–600 | 0.000374 | 0.000563 | 0.000598 |
| Nepheline | 600–850 | 0.000418 | 0.000576 | 0.000600 |

At each wavelength, we extracted a single maximum value for the average polarization along with the corresponding phase angle at each wavelength. The results appear in Figure 10. For the silica and nepheline samples, the lowest value of the polarization maximum occurs near 810 nm. Physically this wavelength corresponds to a local minimum of water absorption [30]. Although all samples were dried in an oven prior to measurement, hygroscopic absorption could still play a role, and this feature near 810 nm may indicate that hygroscopic moisture might be present in the sample data. However, this lowest value of the polarization maximum is in a broad minimum in the silica and nepheline plots, and we also observe that the lowest value of the polarization maximum for the olivine samples is closer to 700 nm, which also casts doubt on the presence of hygroscopic moisture.

In Figure 10, we see that the phase angle of the maximum falls generally within the 100°–120° range predicted by theory, excluding the noise in some of the samples. For the typical indices of refraction of the materials used in this study, the Fresnel equations predict a polarization maximum in this range. We also observe that the phase angle of the linear polarization ratio maximum of these samples is only very weakly correlated with wavelength in some of the materials measured; instead, the phase angle of the maximum appears to be at certain discrete levels. The phase angle of the maximum has some weak correlation with grain size, but the degree to which this is true varies with the material.

For each sample, we also analyzed the impact of the average grain size on the slope of the linear region of the polarization curve for each sample. The results appear in Figure 11. To illustrate the trends, we calculated slopes of the linear region of the polarization curves for four representative wavelengths at even intervals in the middle of the spectral range of the hyperspectral imaging system. Figure 11 shows the slope of the linear region of the sample data for the average grain size of each sifted sample for each material. Plots appear in this Figure for each of the four example wavelengths in the spectral range of the hyperspectral imaging system. Comparing linear and quadratic models, we found that quadratic fits had the lowest RMSE and best $R^2$ values for all samples, with the exception of the Washington Mills olivine for which a linear model was sufficient. In all cases, the $R^2$

goodness of fit ranged between 0.85 and 1.0. However, one notable distinction is the difference in the concavity of the AGSCO nepheline quadratic fit compared to the quadratic fits obtained for the AGSCO olivine and AGSCO silica samples. Material properties or potentially even particle shape may play a role in this. Since nepheline is highly reflective over a broad range of wavelengths with grains that visually exhibit glint and sparkle, there is likely greater surface scattering and less volume scattering and absorption in this material compared to the others used in our study. Also, the impact of grain size on the slope of the linear region appears to be much stronger for olivine, though the order of the relationship was inconsistent between the two olivine samples.

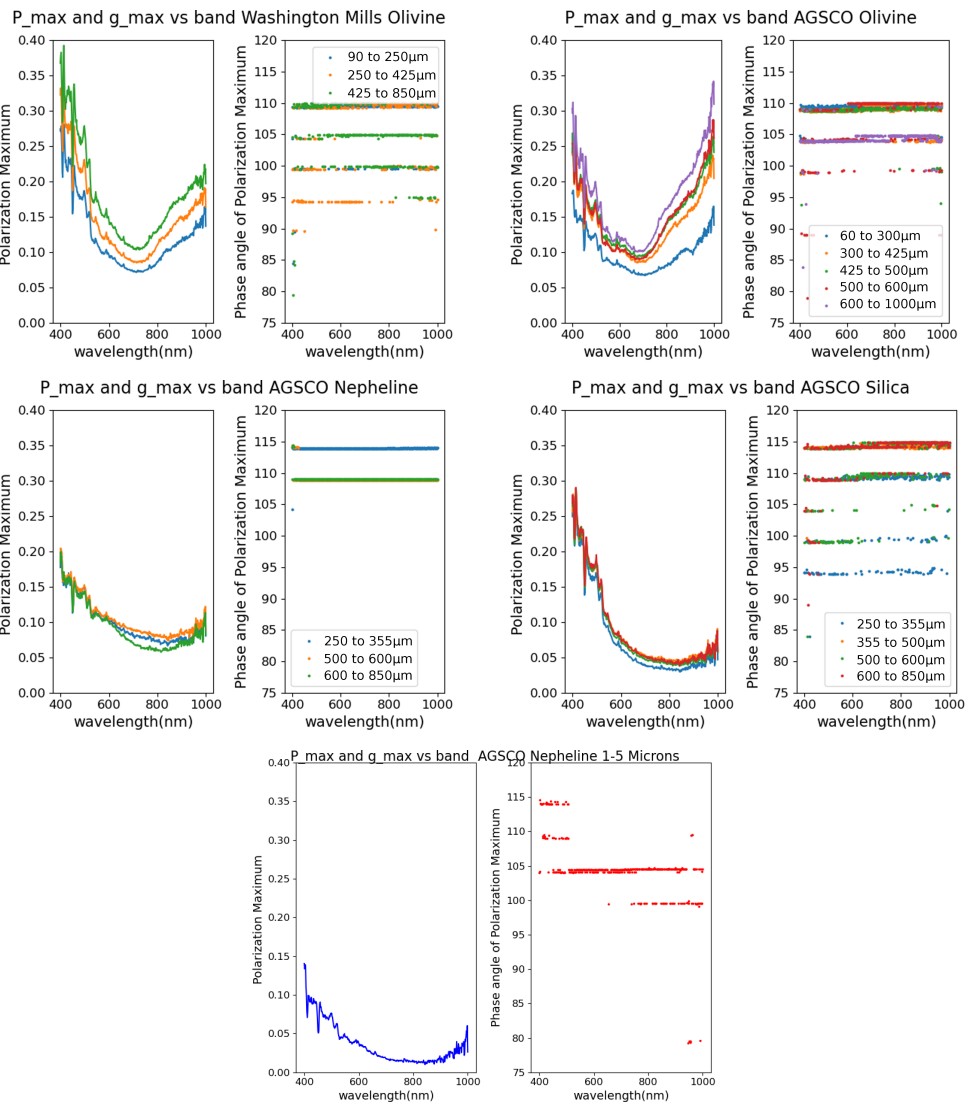

**Figure 10.** Wavelength Dependence of Polarization Maximum and Corresponding Phase Angle After Averaging Sample Lines for (**top left**) Washington Mills Olivine, (**top right**) AGSCO Olivine, (**middle left**) AGSCO Nepheline, (**middle right**) AGSCO Silica, and (**bottom**) AGSCO Nepheline.

To elaborate, as noted in Figure 1, surface scatter will tend to increase polarization while light that has been transmitted through particles and emerges from the particle will tend to be negatively polarized. In particular, the amount of negatively polarized light emerging from particles will depend on the particle diameter because the mean ray path, or average transit length through a particle, should be directly proportional to particle diameter. The theoretical basis of this argument is an equivalent slab model for particle

absorption and transmission discussed by Hapke [7]. In his model, the mean ray path $< D >$ is proportional to the average particle diameter $D$:

$$< D >= \frac{2}{3}\left[n_r^2 - \frac{1}{n_r}(n_r^2 - 1)^{\frac{3}{2}}\right]D \qquad (14)$$

where $n_r$ represents the real part of the index of refraction of the material relative to the surrounding medium (in this case air). Thus, the contribution of negatively polarized light that has been transmitted should be expected to decrease as particles grow larger and extinction increases with the longer average path within the particle. This leads to an increasing polarization maximum as particles increase in size, which is the effect that we observe in Figure 6. The increased peak translates also to a steeper slope in the approximately linear region of the polarization curves between 20°–80°, and thus the observed correlation and increase in the slope in this region as grain size increases; we observe this trend for all but the nepheline sample, which had significantly smaller particle sizes, and for which as noted earlier, the particles are likely dominated by surface scattering rather than volume scattering.

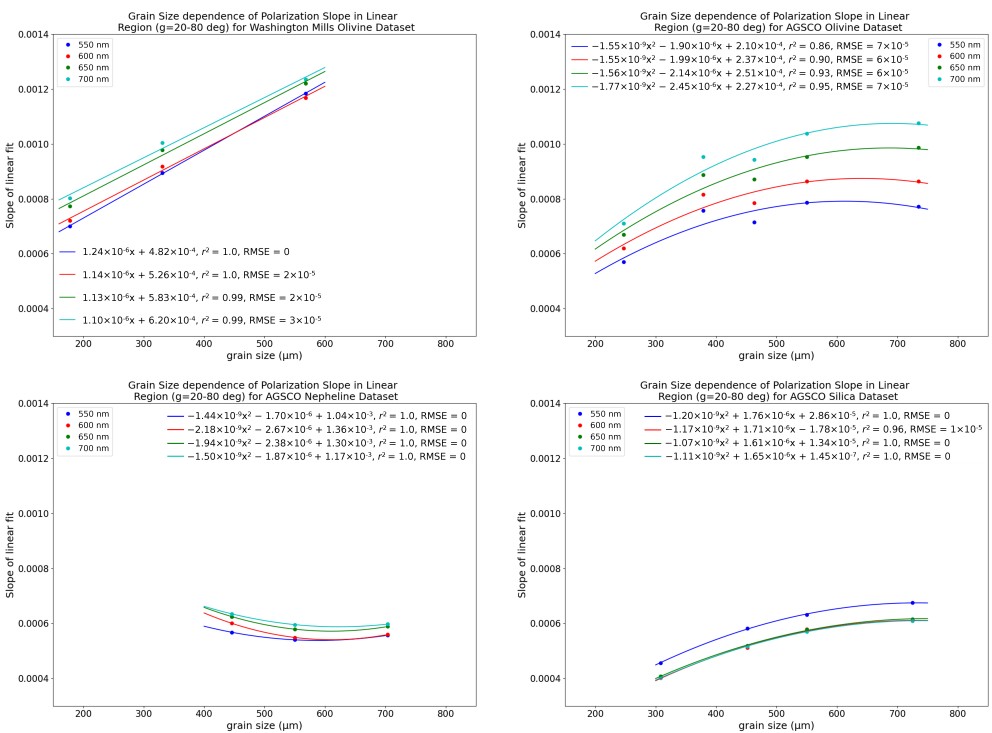

**Figure 11.** Correlation between grain size and slope of linear fit to the polarization data for phase angles between 20 and 80° for example wavelengths for (**top left**) Washington Mills olivine, (**top right**) AGSCO olivine, (**bottom left**) AGSCO nepheline, and (**bottom right**) AGSCO silica.

Returning to the question of why no negative branch of polarization was observed in the geometric optics regime samples, it seems likely that the absence of a negative branch may also be related to a similar mechanism. In backscattered light, especially close to the opposition direction with phase angles less than 20° where the negative branch would be observed if present, light that has been reflected from the surface will be positively polarized, while contributions from light that refract into the particle and then are scattered backward within the volume or from somewhere in the interior surface eventually emerging from the particle, would be negatively polarized. However, in larger particles, this latter contribution from light scattering from the particle volume or interior particle surface and eventually emerging from the particle in a direction toward a sensor positioned at smaller phase angles (<20°) will diminish with longer mean ray paths and therefore greater

extinction. Thus, larger particles will have greater polarization and may not have a negative polarization branch at all, which is what we observed in our samples. In the case of the sample with smaller 1–5 μm nepheline particles, that are in the resonance regime, this contribution may be as small as it is because the material has such strong surface scattering, which is positively polarized, leading to the observed very small scale of the negative branch of linear polarization observed below 20°.

The particulate samples in our analysis were illuminated with a directional source in a controlled laboratory setting. Although this illumination geometry simulates direct sunlight, skylight also contributes to the illumination in outdoor field conditions. Skylight is highly polarized due to Rayleigh scattering, and the polarization of the sky is dependent on the local Sun angles, the distribution of aerosol scatterers, and other atmospheric parameters. For bluer wavelengths, we estimated diffuse illumination in past field experiments by measuring the radiance from a shaded Spectralon$^{TM}$ panel and found that this radiance is approximately 30% compared to that from an unshaded Spectralon panel. For longer wavelengths past 450 nm, this value drops to 3–5%; therefore, we could expect samples illuminated in outdoor environments to be more similar in longer wavelengths to our present laboratory analysis. The effect of skylight illumination, which is also polarized, on the polarimetry of the scene is a complex problem.

## 6. Conclusions

Our results demonstrate a wavelength-dependent relationship between polarization and grain size. However, the grain size relationship appears largely material-dependent. Thus, direct measurement or estimation of the parameters of single scattering albedo, material purity, and index of refraction could expand the usefulness of these results. A rigorous model for the observed relationship is still to be developed; however, we have provided a physical motivation for understanding the results obtained thus far. For most of the materials measured, the slope of the polarization curve increases in the region where the polarization curve is approximately linear between phase angles of 20°–80°. We argued that this can be directly linked to the increasing size of the mean ray path which governs the amount of negatively polarized light that can escape from particle volumes. Past theoretical work by Hapke [7] based on an equivalent slab model of particle absorption and transmission indicates a linear dependence of the mean ray path on typical particle diameter. When typical particle size increases, therefore, we can expect the amount of negatively polarized light available from transmission through particles should decrease, therefore raising the polarization maximum and increasing the slope in the linear region of the polarization curve as average particle size increases. We observed this behavior in all but one material, the nepheline sample, which differed from the rest because scattering from this material was likely dominated by surface scattering, and this behavior likely also led to the much-reduced scale of its negative branch of polarization. For all of the materials, fits between the slope of the polarization data in the linear region between 20° and 80° and the average grain size obeyed either a linear or quadratic relationship with $R^2$ values between 0.855 and 1.00. While we described a plausible mechanism for understanding why these relationships exist, a comprehensive model of the connection between polarization and its correlation with grain size and other geophysical parameters remains a longer-term objective since it has important implications for remote sensing applications. Future studies could explore additional geophysical parameters other than grain size, how this type of data might differ if organic sediments were included, and the impact of skylight or different types of light sources.

**Author Contributions:** Conceptualization, R.M.G. and C.M.B.; methodology, R.M.G. and C.M.B.; software, R.M.G., C.S.L. and C.H.L.; hardware integration, C.S.L.; validation, R.M.G. and C.M.B.; formal analysis, R.M.G. and C.M.B.; investigation, R.M.G. and C.M.B.; resources, C.M.B.; data curation, R.M.G., C.H.L. and C.S.L.; writing—original draft preparation, R.M.G. and C.M.B.; writing—review and editing, R.M.G., C.M.B. and C.H.L.; visualization, R.M.G. and C.M.B.; supervision, C.M.B.; project administration, C.M.B. All authors have read and agreed to the published version of the manuscript.

**Funding:** This research received no external funding.

**Data Availability Statement:** The laboratory data and Python code used to implement these roughness correction models can be found at https://www.doi.org/10.35009/cfccis-8w70 (accessed on 10 July 2023).

**Conflicts of Interest:** The authors declare no conflict of interest.

**Abbreviations**

The following abbreviations are used in this manuscript:

| | |
|---|---|
| Polarization Opposition Effect | POE |
| Zeroth Order Logarithmic Distribution | ZOLD |
| Coherent Backscatter Opposition Effect | CBOE |
| Broad Negative Polarization | BNP |
| Direct Current | DC |
| NIR | Near Infrared |
| nanometers | nm |
| BRDF | Bidirectional Reflectance Distribution Function |
| Region of Interest | ROI |
| Digital Number | DN |
| Signal-to-Noise Ratio | SNR |
| Root Mean Squared Error | RMSE |
| Coefficient of Determintation | $R^2$ |

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
