# Peer review of "The Effect of Grain Size on Hyperspectral Polarization Data of Particulate Material"

_remotesensing, doi:10.3390/rs15143668_

Round 1

Reviewer 1 Report

Figure 1: Why are the lights in different color?

Line 232 – 251: What was the purpose of sifting the samples with significantly larger grain sizes into multiple groups for grain size analysis?

Line 323 – 337: “What is the reason for the much greater increase in polarization maximum with increasing grain size in the Washington Mills olivine compared to the other samples in the study, and why do all samples among the geometric optics regime sized samples show some separation between curves for different grain sizes at phase angles of 45 deg or higher?”

Discussion: would the results be very different when conducting the experiments in outdoor environments? Also, when the materials are animals’ or plants’ cellular, what would the conclusion be like?

Reviewer 2 Report

This paper discussed the relationship between polarization and grain size via a mathematical model and experiments. The basis of this work is Ref.7, and there is no significant modification to this model. Besides, the result that the relationship is material-dependent is not so interesting. This work includes a tedious (too long) background, although its experiments look rich. It cannot be accepted in RS in the current version.

Some minor comments:

1. Please explain or give some evidence for the sentence in Figure 1, i.e., "Multiple scattering: for small angles, coherent backscatter produces negatively polarized light; at larger angles, leads to unpolarized light";

2. Please check Eq. 2; what is the relation between the third term and the first two terms? Multiplication?

3. It is hard to understand Eq. 6 and 7. What does Se, w - Se mean?

4. The background is too tedious. It should be short and clearer.

5. Lines 264-240. What is your light source? Polarized or unpolarized illumination?

6. Eq. 13 is wrong.

7. It seems that the linear polarization ratio in Fig 6 and 7 are too low. How do you make sure this measurement is accurate? 

8. The quality of Fig 11 is too poor.

Reviewer 3 Report

The authors provide experimental results to evaluate the effect of grain size on hyperspectral polarization data of particulate materals. The overall quality is fine. However, the presentation can be further improved. Here are some minor concerns:

1. For figures 7-9, instead of only using different colors for different curves, the authors could add some marks to distinguish different curves more clearly. The curver can be thicker and the font size can be larger.

2. Please highlight the contributions of this work in the end of the first section to make it more clear.

3. As metioned by the authors, the current models/methods may still have some limitations. Please also suggest several promising future directions in the end.

4. Please proofread the manuscript.

Round 2

Reviewer 2 Report

It can be accepted now.